# Who seeks care after intimate partner violence in Cameroon? sociodemographic differences between a hospital and population sample of women

Mark T. Yost[1]*, Kevin J. Blair[1], McKayla Poppens[1], Michelle Mallahi[1‡], Lauren Eyler Dang[2‡], Rasheedat Oke[1‡], Melissa Carvalho[1‡], Georges Alain Etoundi-Mballa[3‡], Alan Hubbard[2‡], Marquise Kouo Ngamby[4‡], Sithombo Maqungo[5,6‡], Kudzai Chironga[5,6‡], Sandra I. McCoy[7‡], Alain Chichom-Mefire[8‡], Catherine Juillard[1], Salome Maswime[5‡], Fanny Nadia Dissak Delon[9]

1 Department of Surgery, Program for the Advancement of Surgical Equity, University of California Los Angeles, Los Angeles, California, United States of America, 2 Division of Biostatistics, School of Public Health, University of California Berkeley, Berkeley, California, United States of America, 3 Department of Disease Epidemic and Pandemic Control, Ministry of Public Health, Yaounde, Cameroon, 4 United Nations Population Fund (UNFPA), Yaounde, Cameroon, 5 Department of Surgery, Division of Global Surgery, University of Cape Town, Cape Town, South Africa, 6 Division of Orthopaedic Surgery, Orthopaedic Trauma Service, University of Cape Town, Cape Town, South Africa, 7 Division of Epidemiology, School of Public Health, University of California Berkeley, Berkeley, California, United States of America, 8 Faculty of Health Sciences, University of Buea, Buea, Cameroon, 9 University of Bamenda, Bamenda, Cameroon

☯ These authors contributed equally to this work.
‡ These authors also contributed equally to this work.
* myost@mednet.ucla.edu

**Data Availability Statement:** All relevant data are within the paper and its Supporting Information files.

## Abstract

### Introduction

Little is known regarding health care seeking behaviors of women in sub-Saharan Africa, specifically Cameroon, who experience violence. The proportion of women who experienced violence enrolled in the Cameroon Trauma Registry (CTR) is lower than expected.

### Methods

We concatenated the databases from the October 2017-December 2020 CTR and 2018 Cameroon Demographic and Health Survey (DHS) into a singular database for cross-sectional study. Continuous and categorical variables were compared with Wilcoxon rank-sum and Fisher's exact test. Multivariable logistic regression examined associations between demographic factors and women belonging to the DHS or CTR cohort. We performed additional classification tree and random forest variable importance analyses.

### Results

276 women (13%) in the CTR and 197 (13.1%) of women in the DHS endorsed violence from any perpetrator. A larger percentage of women in the DHS reported violence from an intimate partner (71.6% vs. 42.7%, p<0.001). CTR women who experienced IPV

**Funding:** M.Y. (first author) received salary support through a grant from the LB Research and Education Foundation dispersed through the H & H Lee Research Program (award number N/A). M.Y. (first author) was supported by a fellowship from the University of California Global Health Institute consortium, funded by the Fogarty International Center of the National Institutes of Health under award number D43TW009343. Research reported in this publication was supported by the Fogarty International Center of the National Institutes of Health under award number R21TW010453 (awarded to Dr. C.J.). The content is solely the responsibility of the authors and does not necessarily represent the official views of the National Institutes of Health.

**Competing interests:** The authors have declared that no competing interests exist.

demonstrated greater university-level education (13.6% vs. 5.0%, p<0.001) and use of liquid petroleum gas (LPG) cooking fuel (64.4% vs. 41.1%, p<0.001). DHS women who experienced IPV reported greater ownership of agricultural land (29.8% vs. 9.3%, p<0.001). On regression, women who experienced IPV using LPG cooking fuel (aOR 2.55, p = 0.002) had greater odds of belonging to the CTR cohort while women who owned agricultural land (aOR 0.34, p = 0.007) had lower odds of presenting to hospital care. Classification tree variable observation demonstrated that LPG cooking fuel predicted a CTR woman who experienced IPV while ownership of agricultural land predicted a DHS woman who experienced IPV.

## Conclusion

Women who experienced violence presenting for hospital care have characteristics associated with higher SES and are less likely to demonstrate factors associated with residence in a rural setting compared to the general population of women experiencing violence.

## Introduction

Intimate partner violence (IPV) affects an estimated 27% of women globally, with greater prevalence in low- and middle-income countries (LMICs) [1]. Globally, women exposed to IPV experience poor health outcomes such as higher rates of anxiety and depression, substance abuse, and suicidality [2,3]. Younger, impoverished women in LMICs have a greater likelihood of experiencing IPV [2,4,5]. Societal norms that accept IPV and lack of accountability for perpetrators also increase the risk of IPV in LMICs [6,7]. Factors associated with lower risk of IPV in LMICs include high socioeconomic status (SES), higher education, and greater women's empowerment [2,6]. IPV is pervasive in Cameroon, as about 37% of women experience physical and/or sexual IPV during their lifetime [8–10]. Approximately 28% of Cameroonian women believe that a husband is justified in beating his wife in certain circumstances [11]. Likewise, the Cameroon Civil Code does not specifically criminalize domestic violence and a husband cannot be legally tried for sexually assaulting his wife [7,8]. One population-based survey in Cameroon found that only eight perpetrators out of a reported 208 episodes of sexual violence faced punishment; possibly reflecting a greater societal indifference towards violence against women [7].

Little is known regarding health care seeking behaviors by women in sub-Saharan Africa (SSA), specifically Cameroon, who experience violence. Female attitudes that accept IPV under certain circumstances hinder care seeking for injuries sustained from IPV [11–13]. Women who experience IPV remain reluctant to seek care due to shame, fear of consequences of reporting violence, and stigma [14,15]. Moreover, economic systems often compel women to remain dependent on male partners by failing to adequately compensate women for their labor, magnifying women's vulnerability and exacerbating the risk for continued violence [13,15]. While the 2018 Cameroon Demographic and Health Survey (DHS) found that women are three times more likely than men to experience physical spousal violence, a recent analysis of the national Cameroon Trauma Registry (CTR) data revealed that women comprise only 22% of all patients who present to the hospital with injuries from interpersonal violence [16,17]. This proportion of women who experienced violence and enrolled in the CTR is lower than expected. This paper seeks to understand the factors that associate with women who present to CTR-affiliated hospitals for evaluation and treatment of violent injuries.

To better understand factors that influence health care seeking behaviors after IPV, we compared Cameroonian women who received hospital-level care for injuries due to violence measured by the CTR to the 2018 Cameroon DHS population sample reporting violence within the past 12 months [16]. In this paper, the CTR cohort represents the population of women who experience violence and present for medical care while the DHS cohort serves as the baseline population of women experiencing violence in Cameroon. Women with higher education and employment are more likely to seek care after experiencing IPV as they have greater economic prospects, potential for financial independence, and have the ability to pay in the Cameroonian fee-for-service model [18–20]. Since access to financial resources and higher education levels greatly empower women, we hypothesized that women of a higher socioeconomic position would be more likely to present to a hospital for care of injuries sustained from IPV [2,6,18,21,22].

## Methods

### Study design

We performed a cross-sectional study using data from the CTR and 2018 Cameroon DHS. We prospectively collected the CTR data at four participating hospitals in Cameroon during a designated study period from October 2017 until January 2020. We compared this dataset to survey data from a nationally representative sample of households acquired by the 2018 Cameroon DHS.

### Setting

From October 2017 through January 2020, four regional hospitals participated in CTR data collection: Laquintinie Hospital of Douala, Limbe Regional Hospital, Pouma Catholic Hospital, and Edea Regional Hospital [23–26]. All four hospitals are located in the Southwest and Littoral governmental regions of Cameroon.

The 2018 Cameroon DHS conducted a nationally representative, publicly available sample of households throughout all ten governmental regions of Cameroon [16]. To make the DHS cohort as similar as possible to the CTR cohort for our study comparison, we only included DHS data from women in surveyed households from the Douala (a city within the Littoral region), Southwest, and Littoral regions of Cameroon. We excluded women from households in other regions of Cameroon to avoid any underlying regional differences that may confound study findings.

### Participants and data sources

We extracted CTR data for women aged 15–49 years who experienced violence and received trauma care at participating CTR hospitals. As part of CTR data collection, women had the opportunity to identify if their injury mechanism was intentional and if the perpetrator was a current or ex-partner. We designated a woman to have experienced violence from all perpetrators if they indicated they suffered an intentional injury regardless of perpetrator (including unknown and omitted perpetrators). Likewise, we defined a woman to have experienced IPV if they indicated that the perpetrator of their intentional injury was either a current or ex-partner.

From the 2018 Cameroon DHS survey, we included women aged 15–49 years who were members of the household in the aforementioned geographic regions where they interviewed and who responded to the domestic violence questions. We selected the age range of 15–49 years to best match the data available between the DHS and CTR databases for comparison.

The DHS reports findings of women who experience violence using the age range of 15–49 years [16]. We extracted data for women who reported experiencing violence within the past 12 months; responses of "often" or "sometimes" in the last 12 months to the physical and sexual violence situations were used. Women were also given the opportunity to identify their relationship to the perpetrator of violence. We defined violence against women by all perpetrators and IPV in the previously described manner. While the DHS asked women who experienced violence about care seeking behaviors, the time frame of these questions inquired if women have "ever" sought care over their lifetime. Since this time frame is not congruent with our analysis parameters (i.e., care seeking behaviors within the past 12 months), those variables were excluded from this analysis.

## Statistical methods and data analysis

We concatenated the databases from the CTR and 2018 Cameroon DHS into a singular database for comparison. Our primary outcome was whether a woman who experienced violence belonged to the CTR or DHS dataset. The CTR represented seeking medical care for violent injuries and the DHS sample represented the background population of Cameroonian women who had experienced violence. The differences and similarities of variable descriptions between the CTR and DHS databases are listed as a (S1 Table). We compared sociodemographic variables between the CTR and DHS samples using descriptive statistics. Our primary analysis focused on women who reported IPV, and our secondary analysis focused on women reporting violence by any perpetrator. Comparisons were also made between IPV and violence from all perpetrators using Stata version 16 [27]. Continuous variables and categorical variables were compared via Wilcoxon rank-sum and Fisher's exact test, respectively. We then used multivariable logistic regression to examine the associations between demographic factors and women belonging to the DHS cohort (adjusted odds ratio [aOR] < 1) or the CTR cohort (aOR > 1). Stata performed listwise deletion of any missing observations on the outcome and predictor variables in the regression analysis. Each regression model variable had less than 10% of missing data, minimizing the significance of listwise deletion.

Finally, we performed additional classification tree and random forest variable importance exploratory analyses using R version 4.1.3 [28]. These methods aimed to determine the ranked importance of variables and their influence on the outcome. A classification tree assigns each variable observation into a specific category by recursively splitting the data into smaller and smaller subgroups [29]. Interpretation of a classification tree consists of following the branches down from the top while examining the split criteria of each level to understand the decision process of the tree. Likewise, random forest variable importance is a machine learning algorithm that creates an ensemble of decision trees, where each tree focuses on different data subsets and a random subset of the variables [30]. The trees are combined to make a final prediction. Variable importance measures determine which variables are most important for making predictions in a random forest. To calculate variable importance, we used permutation importance by observing how the accuracy of the model decreases when a particular variable is removed or permutated randomly. The displayed forests do not display the direction of the association [29,30]. The correlation of these exploratory analyses with regression findings can augment the accuracy of study findings.

## Ethics

Approval to conduct this study was granted by the University of California Los Angeles (IRB #19–000086) institutional review board (IRB), the University of California San Francisco IRB

(#13–12535), and the Cameroon National Ethics Committee (N˚2014/09/496/CE/CNERSH/SP and N˚2018/09/1094/CE/CNERSH/SP). Trained Cameroonian research assistants approached eligible injured patients to obtain informed verbal consent for enrollment in the CTR. The permission for verbal informed consent for adult patients and parents/guardians of minor patients was granted by the IRB. Research assistants used an IRB approved consent script while enrolling patients in the CTR. Parents and/or guardians provided verbal consent for minors under the age of 18 years old. For each patient who provided consent, the research assistant who had the informed consent discussion with the patient and witnessed the consent documented obtaining verbal consent on the patient trauma registry form. The documentation of patient consent was then recorded in the CTR electronic database.

## Results

From October 2017 through January 2020, 2117 women aged 15 years or older presented with any injury type to CTR-affiliated hospitals. Of those, 276 women (13%) aged 15–49 years endorsed experiencing violence from any type of perpetrator. Out of 1509 women residing in the Southwest, Douala, and Littoral regions surveyed by the 2018 Cameroon DHS, 197 (13.1%) women endorsed experiencing violence from any perpetrator within the past 12 months. Compared to the CTR cohort, a larger percentage of women in the DHS data (71.6% vs. 42.7%, p<0.001) reported violence from an intimate partner (Table 1).

**Table 1. Demographics of women who experienced violence in the previous 12 months (DHS) compared with women presenting with injuries from violence (CTR), all perpetrators.**

| | | DHS N = 197 | CTR N = 276 | |
|---|---|---|---|---|
| | | Percentage (n) | Percentage (n) | p-value |
| Age (median, IQR) | | 30 (24–35) | 29 (24–36) | 0.81 |
| Age groups | | | | 0.11 |
| | Age 15–19 | 14.7 (29) | 10.5 (29) | |
| | Age 20–29 | 31.5 (62) | 40.6 (112) | |
| | Age 30–39 | 40.6 (80) | 33.7 (93) | |
| | Age 40–49 | 13.2 (26) | 15.2 (42) | |
| Urban residence | | 86.3 (170) | 87.3 (241) | 0.78 |
| Own cellphone | | 94.9 (187) | 95.7 (264) | 0.46 |
| | Missing | 0 | 0.7 (2) | |
| Currently employed | | 59.4 (117) | 50.0 (138) | 0.048* |
| | Missing | 0 | 1.1 (3) | |
| Own agricultural land | | 31.5 (62) | 9.8 (27) | <0.001** |
| | Missing | 0 | 2.2 (6) | |
| Own home | | 13.7 (27) | 17.0 (47) | 0.008** |
| | Missing | 0 | 3.6 (10) | |
| University-level education | | 7.6 (15) | 13.0 (36) | <0.001** |
| | Missing | 0 | 8.7 (24) | |
| LPG cooking fuel | | 43.7 (86) | 66.3 (183) | <0.001** |
| | Missing | 0 | 6.9 (19) | |
| Violence perpetrator intimate partner | | 71.6 (141) | 42.7 (118) | <0.001** |

DHS = 2018 Cameroon Demographic and Health Survey, CTR = Cameroon Trauma Registry, IQR = interquartile range, LPG = liquid petroleum gas.

* = p-value less than 0.05.

** = p-value less than 0.01.

**Table 2. Demographics of women who experienced violence in the previous 12 months (DHS) compared with women presenting with injuries from violence (CTR), intimate partner perpetrators.**

| | | DHS N = 141 | CTR N = 118 | |
|---|---|---|---|---|
| | | Percentage (n) | Percentage (n) | p-value |
| Age (median, IQR) | | 31 (27–36) | 29 (25–35) | 0.06 |
| Age groups | | | | 0.21 |
| | Age 15–19 | 2.8 (4) | 2.5 (3) | |
| | Age 20–29 | 36.9 (52) | 50.0 (59) | |
| | Age 30–39 | 45.4 (64) | 35.6 (42) | |
| | Age 40–49 | 14.9 (21) | 11.9 (14) | |
| Urban residence | | 83.7 (118) | 91.5 (108) | 0.06 |
| Own cellphone | | 94.3 (133) | 95.8 (113) | 0.17 |
| | Missing | 0 | 1.7 (2) | |
| Currently employed | | 60.3 (85) | 53.4 (63) | 0.26 |
| | Missing | 0 | 0.9 (1) | |
| Own agricultural land | | 29.8 (42) | 9.3 (11) | <0.001** |
| | Missing | 0 | 0.9 (1) | |
| Own home | | 17.7 (25) | 18.6 (22) | 0.09 |
| | Missing | 0 | 3.4 (4) | |
| University-level education | | 5.0 (7) | 13.6 (16) | <0.001** |
| | Missing | 0 | 6.8 (8) | |
| LPG cooking fuel | | 41.1 (58) | 64.4 (76) | <0.001** |
| | Missing | 0 | 7.6 (9) | |

DHS = 2018 Cameroon Demographic and Health Survey, CTR = Cameroon Trauma Registry, IQR = interquartile range, IPV = intimate partner violence, LPG = liquid petroleum gas.

* = p-value less than 0.05.

** = p-value less than 0.01.

When specifically examining the two cohorts reporting IPV (Table 2), a greater proportion of women presenting to CTR-affiliated hospitals were aged 20–29 years (50.0% vs. 36.9%, p = 0.04). The CTR cohort also demonstrated a greater percentage of university-level education (13.6% vs. 5.0%, p<0.001) and use of liquid petroleum gas (LPG) cooking fuel (64.4% vs. 41.1%, p<0.001) compared to the DHS cohort. Likewise, the DHS cohort demonstrated greater proportion of women reporting ownership of agricultural land (29.8% vs. 9.3%, p<0.001).

Multivariable logistic regression analysis found that women who experienced IPV whose households used LPG cooking fuel (aOR 2.55, p = 0.002) had greater odds of belonging to the CTR cohort. Conversely, women who were 30–39 years old (aOR 0.46, p = 0.012) and whose household owned agricultural land (aOR 0.34, p = 0.007) had lower odds of being from the CTR cohort, suggesting lower odds of presenting for hospital-level care for injuries due to IPV.

Among the women reporting IPV, classification tree variable observation demonstrated that use of LPG cooking fuel predicted that a woman who experienced IPV originated from the CTR population (Fig 1). Furthermore, ownership of agricultural land significantly predicted that a woman who experienced IPV originated from the DHS population. The predicting variables of the classification tree are use of LPG cooking fuel, age, ownership of agricultural land, home ownership, and current employment status. For instance, a 25-year-

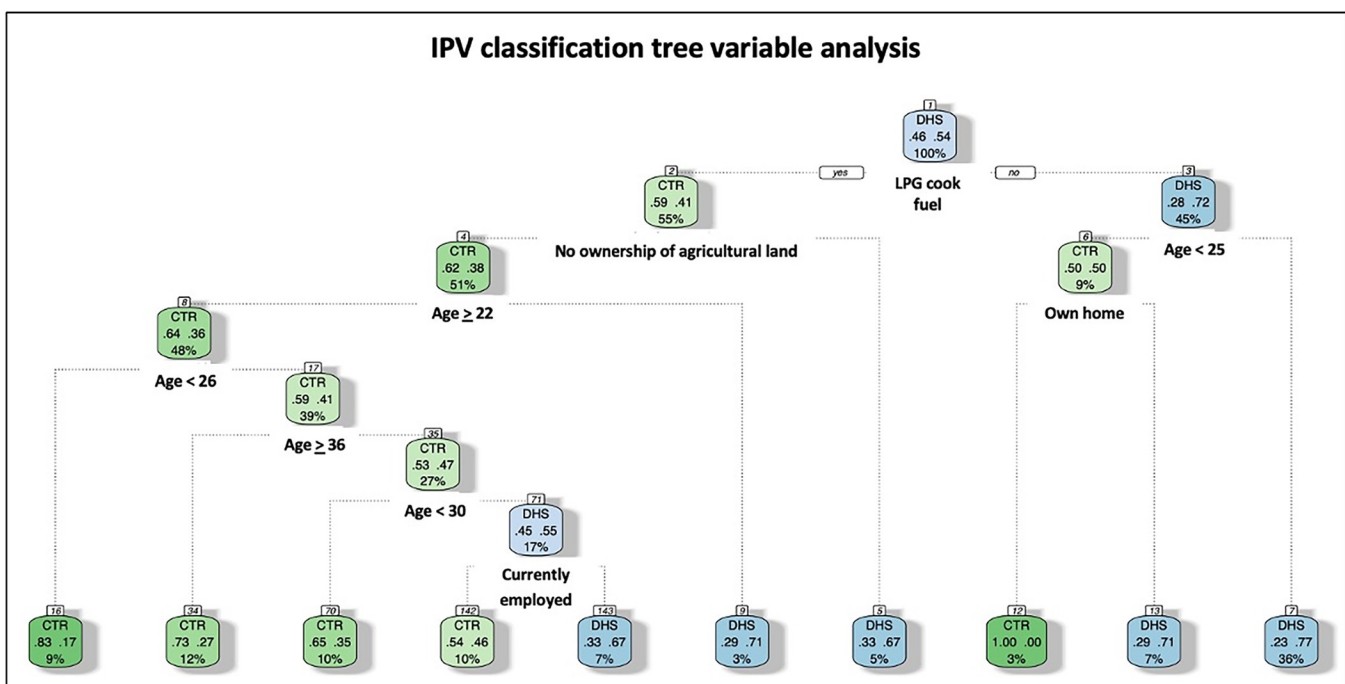

**Fig 1. IPV classification tree variable analysis.** On the classification tree, a branch to the right indicates a "yes" response to the bolded prompt while a branch to the left indicates a "no" response. IPV = intimate partner violence, LPG = liquid petroleum gas, DHS = 2018 Cameroon Demographic and Health Survey, CTR = Cameroon Trauma Registry.

old Cameroonian woman who uses LPG cooking fuel and does not own agricultural land is predicted to belong to the CTR cohort and present to a hospital for injuries due to IPV. Women presenting to hospitals for treatment of IPV injuries tend to be younger, as women older than 25 years old who do not use LPG cooking fuel were more likely to belong to the DHS cohort (Fig 1).

When comparing cohorts, use of LPG cooking fuel by women who experienced IPV is the most important variable for predicting if a woman will present to a hospital for injuries due to IPV (Fig 2). In other words, removing LPG cooking fuel variable from analysis will result in the greatest loss of model accuracy. Ownership of agricultural land and age represent the next two most important prediction variables, respectively.

## Discussion

Cameroonian women who experienced IPV and presented to CTR-affiliated hospitals for treatment of injuries were more likely than those in the DHS to have a university-level education and use LPG cooking fuel. Additionally, women presenting to CTR hospitals were less likely to own agricultural land compared to the women reporting recent violence in the DHS dataset. These findings suggest that women of a higher socioeconomic position living in closer proximity to urban areas may facilitate presentation for hospital-level care after experiencing IPV in Cameroon. These findings could partially explain why there are higher proportions of women reporting violence in the community compared to those receiving care for injuries in the hospital. Since nearly 40% of Cameroonian women experience IPV, violence against women remains a significant issue in Cameroon [8–10]. While this study serves as an initial analysis of the care seeking behaviors of women who experience violence, Cameroonian

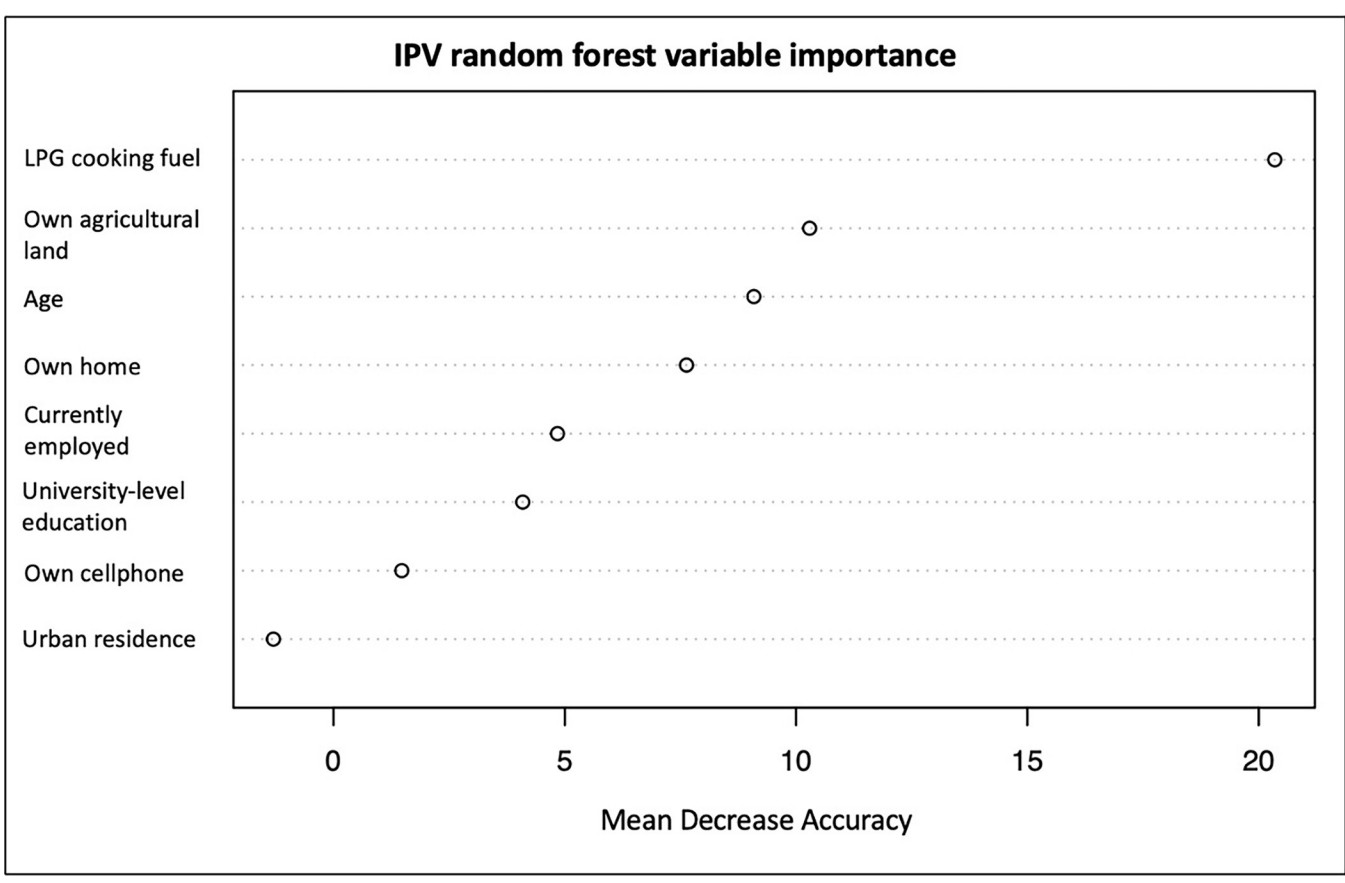

**Fig 2. IPV random forest variable importance.** The figure indicates that excluding the LPG cooking fuel variable from the random forest model would result in the greatest loss of accuracy. Other variables of importance for maintaining model accuracy include ownership of agricultural land and age. These variables are of the greatest importance when predicting belonging to the CTR or DHS cohorts. LPG = liquid petroleum gas.

practitioners can incorporate these findings when screening for violence against women to provide survivor-centered care and referrals to additional resources.

Though the initial analysis supported the hypothesis that women with higher education levels were more likely to present to a hospital for care of injuries due to IPV, the association between university-level education and the CTR cohort did not persist in multivariable regression. While higher education is a marker of women's empowerment, additional factors such as employment may actually increase risk of violence against women, which correlates with our findings that the DHS cohort (i.e., the proxy background population) women who experienced violence from all perpetrators demonstrated greater proportion of employment (Table 1) [6]. The association between markers of female empowerment and violence against women may echo literature that attributes these findings to male backlash, a hostile male reaction to female empowerment in a gender role-based society [31]. Furthermore, a study of long-term effects of colonialism on women in Cameroon determined that while colonialism empowered women to access paid employment, it left women highly susceptible to experiencing domestic violence [32].

Regarding the hypothesis that women with factors corresponding to a higher SES would be more likely to present to the hospital with injuries due to IPV, multivariable regression demonstrated that ownership of agricultural land correlated with lower likelihood of presentation to a hospital. It remains unclear if ownership of agricultural land directly correlates with SES. Prior *EconomicClusters* model analysis of Cameroon DHS data shows that land ownership is

an asset associated with both poor and rich households [33,34]. Nevertheless, those living in rural settings more often own agricultural land [33,34]. Since rural settings are more likely to be greater distances from hospital facilities, LMIC rural households remain vulnerable to poor health outcomes as modeled by the Three Delays framework, specifically delay two (i.e., reaching/arriving at health care) [35]. Thus, women presenting to CTR-affiliated hospitals may demonstrate lower proportion of agricultural land ownership because they live in a mixed urban-rural setting. It is important note that there was no significant difference between urban residence proportions of women who experienced IPV in both cohorts. However, multivariable regression determined that women who used LPG cooking fuel had greater odds of presenting to hospital with injuries sustained from IPV. Additionally, random forest variable importance conveys LPG cooking fuel type to be the most important variable for predicting hospital presentation for injuries sustained from IPV. In Cameroon, LPG is the most expensive cooking fuel type, followed by kerosene, charcoal/coal, and wood/sawdust, respectively [34,36]. Use of LPG is associated with higher levels of household income and asset ownership [36]. *EconomicCluster* analysis reveals LPG to be an indicator of asset rich clusters while charcoal cooking fuel represents the middle class [33]. This finding appears to align with our hypothesis, as women who experienced violence using LPG were more likely to belong to the CTR dataset and present for hospital care. Additionally, this finding correlates with prior *EconomicCluster* analysis that associated urban poor and rural poor clusters with increased likelihood of injuries due to violence in Cameroonian men and women [17].

While similar proportions of women and men report violence in the 2018 Cameroon DHS, a significantly lesser proportion of women report injuries due to IPV in the CTR [16,17]. It has long been suspected that the true prevalence of IPV globally remains underreported due to the sensitive nature of the topic [37,38]. Measurement methods that emphasize respondent anonymity (i.e., indirect method) often reveal greater rates of IPV compared to direct self-report [37]. The DHS and CTR databases utilize self-report methods and likely underreport the prevalence of violence against women in Cameroon. However, the CTR may demonstrate greater amounts of underreporting by women since they need to first present to the hospital and later disclose the cause and perpetrator of their injuries. According to the Three Delays framework, women encounter significant barriers in reaching, arriving at, and receiving health care [35]. The fear and shame often associated with violence, especially IPV, possibly diminishes care seeking behavior by exacerbating the delay in health care presentation and may hinder the truthful identification of the perpetrator [14,38]. Lastly, the difference in the proportion of IPV between the DHS and CTR cohort compared to violence from all perpetrators may stem from the fact that the DHS contains more questions specific to violence from a partner rather than other perpetrators.

This study contains several limitations. First, the analysis is restricted to variables that are matching between the DHS and CTR databases. Among variables available in both datasets, certain questions were asked differently between the two surveys, as detailed in the Methods section (S1 Table). Variations in question wording for certain variables may have contributed to differences in results between the two samples. The use of participant age range of 15–49 years is consistent with other studies utilizing DHS domestic violence survey data [39]. We recognize that the factors influencing injury circumstances and access to access vary greatly across different age groups. However, our small sample size limited any sub-group analyses based on age. The CTR from October 2017 until January 2020 did not record marital status, a known risk factor for IPV, however the current CTR intake forms have been revised to record marital status for future data analysis. The DHS dataset may be influenced by recall bias, as the survey inquired if women experienced violence within the past 12 months. When examining violence against women, the sensitive nature of discussing such topics increases the broader

**Table 3. Multivariable logistic regression of women who experienced IPV.**

|  |  | Adjusted odds ratio | 95% CI | p-value |
|---|---|---|---|---|
| Age group |  |  |  |  |
|  | 15–19 | 1.43 | 0.27–7.61 | 0.67 |
|  | 20–29 | Reference[#] | – | – |
|  | 30–39 | 0.46 | 0.25–0.84 | 0.012* |
|  | 40–49 | 0.62 | 0.26–1.46 | 0.28 |
| Own agricultural land |  | 0.34 | 0.16–0.75 | 0.007** |
| University-level education |  | 2.32 | 0.86–6.25 | 0.10 |
| LPG cooking fuel |  | 2.55 | 1.43–4.55 | 0.002** |

95% CI = 95% confidence interval, LPG = liquid petroleum gas, IPV = intimate partner violence.

[#]Note: The age group with the most observations was chosen as the reference level.

* = p-value less than 0.05.

** = p-value less than 0.01.

risk of reporting bias. Additionally, data missingness in the CTR dataset reflects the limitations of real-world trauma registry data collection. We display the data missingness in Table 2 for all variables included in our regression model shown in Table 3. Each variable included in the regression model had less than 10% of missing data, minimizing the impact of listwise deletion. Using the CTR as a proxy for the population of injured women who experienced violence presenting to a hospital is limited by small sample size and selection bias. Similarly, using the DHS dataset as a proxy for baseline prevalence of violence against women in Cameroon is limited by small sample size and the lack of information about care seeking behavior of the DHS cohort. The cross-sectional methodology cannot establish causative relationships between variables and outcomes. Furthermore, the interpretation of these findings is related to whether the population who presents for hospital care is different than the population who experiences violence in general. The analysis is limited by substituting the population of women experiencing IPV in the DHS for the unobserved population of women who experienced IPV and did not seek care. We do not know if women in DHS sought care for injuries caused by violence or if women in the DHS appear in the CTR dataset. Finally, other factors such as mass media exposure does not represent a significant confounder as the majority of Cameroonians have access to modern cellphone technology [23,40].

While these limitations restrict the generalizability and obscure the policy implications of these findings, this study can inform future research that directly measures the factors that associate with care-seeking behaviors of women who experience violence. These findings remain informative for stakeholders, such as health care providers and patients, about the factors that associate with care-seeking behaviors of women who experience violence. Future studies could leverage a recent expansion of CTR-affiliated facilities from four to ten hospitals to collect data from a larger sample of women and further elucidate our findings. For instance, a prospective study with a planned control group could utilize an *EconomicCluster* analysis to better delineate the socioeconomic factors that contribute to care seeking behaviors among women who experience IPV. Moreover, though a qualitative approach was outside the scope of this cross-sectional quantitative study, future research conducting qualitative interviews of women who experience violence in Cameroon would provide a more nuanced understanding of care seeking facilitators and barriers among this patient population. The complex natures of IPV requires multiple research methods to develop an in-depth understand the contributing factors and influence context-specific mitigation interventions. Additional interviews of

stakeholders such as community leaders, health care providers, and even former perpetrators of violence against women could reveal important social perceptions of IPV in Cameroonian society. These qualitative findings could inform future IPV screening and prevention efforts and influence post-violence recovery interventions. Finally, dissemination of such research findings would be critical to raising societal awareness of the public health impact of IPV and building sociopolitical support for future mitigation interventions. A dissemination plan that leverages the CTR network of affiliated hospitals throughout Cameroon could publicize findings among providers and patients.

## Conclusions

Women who experienced violence presenting to the hospital have characteristics associated with higher SES and are less likely to demonstrate factors associated with residence in a rural setting compared to the general population of women experiencing violence. There remains a critical need to further understand the public health problem of IPV in Cameroon. Future studies employing both quantitative and qualitative methods can elucidate an in-depth understanding of the facilitators and barriers that contribute to IPV. These findings can guide future IPV prevention and post-IPV recovery interventions in the Cameroonian context.

## Supporting information

**S1 Checklist.**
(DOCX)

**S1 Table. Comparison of variables in CTR and DHS datasets.** While the concatenated CTR and DHS databases had several similarities between variables, each database defined certain variables slightly differently. This table explains the differences in variables and highlights how the variables were considered during data analysis. CTR = Cameroon Trauma Registry, DHS = Demographic and Health Survey, LPG = liquid petroleum gas.
(DOCX)

**S1 Data. Data used in study analysis.** Deidentified data made available to the *PLOS Global Public Health* as requested. CTR = Cameroon Trauma Registry, DHS = Demographic and Health Survey, LPG = liquid petroleum gas; outcome = database from which study participant originated; violence_12mos_partner = participant experienced violence from intimate partner in past 12 months; violence_12mos = participant experienced violence from any perpetrator in past 12 months; age = participant age; age group = participant age group; urban = participant urban or rural location of residence; cell_combined = participant ownership of cellphone; work_current = participant currently employed; agland_new = participant ownership of agricultural land; ownhome = participant ownership of home; education3 = participant university level of education; cookfuel_lpg = participant used LPG cooking fuel.
(XLSX)

## Acknowledgments

We acknowledge the efforts of the dedicated Cameroonian trauma registry staff at each participating CTR hospital who were involved in data collection and entry. We also acknowledge and appreciate the support of the Cameroonian Ministry of Public Health. The abstract for this manuscript was presented by author Fanny Dissak-Delon at the American Public Health Association (APHA) Conference in Boston, MA, USA on November 8, 2022.

## Author Contributions

**Conceptualization:** Kevin J. Blair, Michelle Mallahi, Lauren Eyler Dang, Georges Alain Etoundi-Mballa, Marquise Kouo Ngamby, Sithombo Maqungo, Kudzai Chironga, Sandra I. McCoy, Alain Chichom-Mefire, Catherine Juillard, Salome Maswime, Fanny Nadia Dissak Delon.

**Data curation:** Mark T. Yost, Kevin J. Blair, McKayla Poppens, Michelle Mallahi, Rasheedat Oke, Melissa Carvalho, Alan Hubbard.

**Formal analysis:** Mark T. Yost, Kevin J. Blair, McKayla Poppens, Michelle Mallahi, Alan Hubbard, Fanny Nadia Dissak Delon.

**Funding acquisition:** Alain Chichom-Mefire, Catherine Juillard.

**Investigation:** Kevin J. Blair, Alan Hubbard.

**Methodology:** Mark T. Yost, Kevin J. Blair, McKayla Poppens, Alan Hubbard, Fanny Nadia Dissak Delon.

**Project administration:** Rasheedat Oke, Melissa Carvalho.

**Supervision:** Alain Chichom-Mefire, Catherine Juillard, Salome Maswime, Fanny Nadia Dissak Delon.

**Writing – original draft:** Mark T. Yost.

**Writing – review & editing:** Mark T. Yost, Kevin J. Blair, McKayla Poppens, Lauren Eyler Dang, Rasheedat Oke, Melissa Carvalho, Georges Alain Etoundi-Mballa, Alan Hubbard, Marquise Kouo Ngamby, Sithombo Maqungo, Kudzai Chironga, Sandra I. McCoy, Alain Chichom-Mefire, Catherine Juillard, Salome Maswime, Fanny Nadia Dissak Delon.

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
