## [Decision Letter · Decision Letter 0]

22 Jan 2024

PGPH-D-23-02019

Who seeks care after intimate partner violence in Cameroon? Sociodemographic differences between a hospital and population sample of women

Dear Dr. Mark Thomas Yost,

Thank you for submitting your manuscript to PLOS Global Public Health. After careful consideration, we feel that it has merit but does not fully meet PLOS Global Public Health’s publication criteria as it currently stands. Therefore, we invite you to submit a revised version of the manuscript that addresses the points raised during the review process.

We look forward to receiving your revised manuscript.

Kind regards,

Jayanta Kumar Bora,PhD

Academic Editor

Journal Requirements:

1. You indicated that you had ethical approval for your study. In your Methods section, please ensure you have also stated whether you obtained consent from parents or guardians of the minors included in the study or whether the research ethics committee or IRB specifically waived the need for their consent."""

2. Please include a complete copy of PLOS’ questionnaire on inclusivity in global research in your revised manuscript. Our policy for research in this area aims to improve transparency in the reporting of research performed outside of researchers’ own country or community. The policy applies to researchers who have travelled to a different country to conduct research, research with Indigenous populations or their lands, and research on cultural artefacts. The questionnaire can also be requested at the journal’s discretion for any other submissions, even if these conditions are not met.  Please find more information on the policy and a link to download a blank copy of the questionnaire here: https://journals.plos.org/globalpublichealth/s/best-practices-in-research-reporting. Please upload a completed version of your questionnaire as Supporting Information when you resubmit your manuscript.

Additional Editor Comments (if provided):

Reviewers' comments:

Reviewer's Responses to Questions

**Comments to the Author**

1. Does this manuscript meet PLOS Global Public Health’s publication criteria? Is the manuscript technically sound, and do the data support the conclusions? The manuscript must describe methodologically and ethically rigorous research with conclusions that are appropriately drawn based on the data presented.

Reviewer #1: Yes

Reviewer #2: Yes

2. Has the statistical analysis been performed appropriately and rigorously?

Reviewer #1: Yes

Reviewer #2: Yes

3. Have the authors made all data underlying the findings in their manuscript fully available (please refer to the Data Availability Statement at the start of the manuscript PDF file)?

Reviewer #1: Yes

Reviewer #2: Yes

4. Is the manuscript presented in an intelligible fashion and written in standard English?

Reviewer #1: Yes

Reviewer #2: Yes

5. Review Comments to the Author

Reviewer #1: I think PLOS criteria has been met, but the larger issue of VAW needs to be acknowledged and how this study contributes to the larger goal could be included.

Additional comments: a para could be added in the introduction explaining why interviews with survivors were not conducted, and why it is important to interview them. The paper mentions that qualitative approach was not taken but does not explain why it could not be. this gap should be stated clearly stated. Mention of stakeholders for VAW mitigation could be included and a dissemination plan for reaching them outlined.

Reviewer #2: The paper was adequately motivated and no doubt has a valid research gap which has been addressed. However, there are extant literature on IPV that should be reviewed to provide a basis for the hypothesis "that women with higher SES and education levels would be more likely to present to a hospital for care of injuries sustained from IPV". Also, at best the findings are mere "associations". Whether policy implication can be drawn from such findings is unclear. This should interest stakeholders.

Additionally, on lines 107-108 “ We excluded women 108 from households in other regions of Cameroon” Can they authors provide rationale for the exclusion? And other similar eligibility criteria used in the selection of the final sample? Further, on lines 142-143 “142 Any missing observations on the outcome and predictor variables underwent listwise 143 deletion in the regression”. Can they authors mention such variables and how "significant" were their omission?

Finally, does the "conclusion that Women who experienced violence presenting to the hospital have characteristics associated with higher SES and are less likely to demonstrate factors associated with residence in rural setting compared to the general population of women experiencing violence" free from self selection bias? Could factor such as exposure to mass media confound the findings?

6. PLOS authors have the option to publish the peer review history of their article (what does this mean?). If published, this will include your full peer review and any attached files.

**Do you want your identity to be public for this peer review?** For information about this choice, including consent withdrawal, please see our Privacy Policy.

Reviewer #1: No

Reviewer #2: **Yes: **Atsiya PIUS AMOS

---

## [Decision Letter · Decision Letter 1]

10 May 2024

PGPH-D-23-02019R1

Who seeks care after intimate partner violence in Cameroon? Sociodemographic differences between a hospital and population sample of women

Dear Dr. Yost,

Thank you for submitting your manuscript to PLOS Global Public Health. After careful consideration, we feel that it has merit but does not fully meet PLOS Global Public Health’s publication criteria as it currently stands. Therefore, we invite you to submit a revised version of the manuscript that addresses the points raised during the review process.

The manuscript has been evaluated by one reviewer, and their comments are available below.

The reviewer has raised multiple minor concerns, specifically they recommend that you improve the details of the methods section and expand on the results and discussion. Could you please carefully revise the manuscript to address all comments raised?

We look forward to receiving your revised manuscript.

Kind regards,

Johanna Pruller, Ph.D.

PLOS Staff Editor

Journal Requirements:

1. You indicated that you had ethical approval for your study. In your Methods section, please ensure you have also stated whether you obtained consent from parents or guardians of the minors included in the study or whether the research ethics committee or IRB specifically waived the need for their consent."""

2. Please include the following request in the decision letter, and ping me with follow up. “Please include a complete copy of PLOS’ questionnaire on inclusivity in global research in your revised manuscript. Our policy for research in this area aims to improve transparency in the reporting of research performed outside of researchers’ own country or community. The policy applies to researchers who have travelled to a different country to conduct research, research with Indigenous populations or their lands, and research on cultural artefacts. The questionnaire can also be requested at the journal’s discretion for any other submissions, even if these conditions are not met.  Please find more information on the policy and a link to download a blank copy of the questionnaire here: https://journals.plos.org/globalpublichealth/s/best-practices-in-research-reporting. Please upload a completed version of your questionnaire as Supporting Information when you resubmit your manuscript.

Additional Editor Comments (if provided):

Reviewers' comments:

Reviewer's Responses to Questions

**Comments to the Author**

1. If the authors have adequately addressed your comments raised in a previous round of review and you feel that this manuscript is now acceptable for publication, you may indicate that here to bypass the “Comments to the Author” section, enter your conflict of interest statement in the “Confidential to Editor” section, and submit your "Accept" recommendation.

Reviewer #3: (No Response)

2. Does this manuscript meet PLOS Global Public Health’s publication criteria? Is the manuscript technically sound, and do the data support the conclusions? The manuscript must describe methodologically and ethically rigorous research with conclusions that are appropriately drawn based on the data presented.

Reviewer #3: Yes

3. Has the statistical analysis been performed appropriately and rigorously?

Reviewer #3: Yes

4. Have the authors made all data underlying the findings in their manuscript fully available (please refer to the Data Availability Statement at the start of the manuscript PDF file)?

Reviewer #3: Yes

5. Is the manuscript presented in an intelligible fashion and written in standard English?

Reviewer #3: Yes

6. Review Comments to the Author

Reviewer #3: First of all, I would like to congratulate you for the work you have done and in particular for trying to focus on a problem of global magnitude such as violence against women, specifically Intimate Partner Violence (IPV). Unfortunately, in the context of the study, the approach taken is one of the possible ones, although it is the tip of the iceberg of the problem. The study is important in itself because of what it shows, but even more so because of what it is not possible to show. I think the authors are aware of the latter, and it would be of interest if it were shown in the article.

I will make specific comments on the following article:

Methods.

Explain on the basis of which you have chosen the 15-49 age range, and what this decision entails. On the one hand, you include minors, 15-18 years old, young women, 18-35 years old, and adult women, 35-49 years old. The circumstances of minor women will be different from those of young or adult women. In this regard, I think it is important to explain why this decision was made, and to think about whether to add additional information where appropriate in this regard. It would be desirable to find information on this in the limitations section of the study.

Results and discusión

Incorporate a bibliographic reference that contextualizes in depth the importance of the LPG indicator. I suggest a bibliographic reference, by way of example, but consider incorporating a more appropriate one.

Household Determinants of Liquified Petroleum Gas (LPG) as a Cooking Fuel in South West Cameroon. Pope D, Bruce N, Higgerson J, Hyseni L, Ronzi S, Stanistreet D, MBatchou B, Puzzolo E. Ecohealth. 2018 Dec;15(4):729-743. doi: 10.1007/s10393-018-1367-9. Epub 2018 Oct 1.

It is positive that you are providing information in relation to new lines of research. Specifically, they refer to research in which qualitative methodology would be used. In this case, I suggest that you do not only talk about a qualitative research technique, such as the interview, but also allude to the importance of seeking answers to questions that can be answered using another type of methodology, such as qualitative. Indeed, collecting qualitative information among women who have suffered IPV, health professionals, state security forces, social workers, workers of non-governmental organizations, and even, as you say, among the perpetrators of IPV, would be essential to gain an in-depth understanding of the problem you are addressing. It would also be interesting to comment on the need to develop research to know the social perceptions of the population of Cameroon in relation to IPV, as well as studies aimed at raising awareness among the population in relation to this social and public health problem. Personally, I encourage you to develop these future lines of research.

Conclusions.

I suggest that the final sentences of the conclusions be along the lines of reinforcing issues raised in the manuscript, beyond the specific study, in relation to the need to develop more research studies on IPV in the context of Cameroon, in order to gain an in-depth understanding of this problem, which is a global problem.

7. PLOS authors have the option to publish the peer review history of their article (what does this mean?). If published, this will include your full peer review and any attached files.

**Do you want your identity to be public for this peer review?** For information about this choice, including consent withdrawal, please see our Privacy Policy.

Reviewer #3: No

---

## [Decision Letter · Decision Letter 2]

5 Jun 2024

Who seeks care after intimate partner violence in Cameroon? Sociodemographic differences between a hospital and population sample of women

PGPH-D-23-02019R2

Dear Dr. Yost,

We are pleased to inform you that your manuscript 'Who seeks care after intimate partner violence in Cameroon? Sociodemographic differences between a hospital and population sample of women' has been provisionally accepted for publication in PLOS Global Public Health.

Best regards,

Julia Robinson

Executive Editor

Reviewer Comments (if any, and for reference):

Reviewer's Responses to Questions

**Comments to the Author**

1. If the authors have adequately addressed your comments raised in a previous round of review and you feel that this manuscript is now acceptable for publication, you may indicate that here to bypass the “Comments to the Author” section, enter your conflict of interest statement in the “Confidential to Editor” section, and submit your "Accept" recommendation.

Reviewer #3: All comments have been addressed

2. Does this manuscript meet PLOS Global Public Health’s publication criteria? Is the manuscript technically sound, and do the data support the conclusions? The manuscript must describe methodologically and ethically rigorous research with conclusions that are appropriately drawn based on the data presented.

Reviewer #3: Yes

3. Has the statistical analysis been performed appropriately and rigorously?

Reviewer #3: Yes

4. Have the authors made all data underlying the findings in their manuscript fully available (please refer to the Data Availability Statement at the start of the manuscript PDF file)?

Reviewer #3: Yes

5. Is the manuscript presented in an intelligible fashion and written in standard English?

Reviewer #3: Yes

6. Review Comments to the Author

Reviewer #3: The authors have incorporated responses to the comments made. In relation to the article they have written, I want to highlight the importance of the problem they address, and the complexity of studying this problem in the context in which they have carried it out. I encourage the authors to continue this line of work that is so necessary.

7. PLOS authors have the option to publish the peer review history of their article (what does this mean?). If published, this will include your full peer review and any attached files.

**Do you want your identity to be public for this peer review?** For information about this choice, including consent withdrawal, please see our Privacy Policy.

Reviewer #3: No
